# Do Attitudes towards Work or Work Motivation Affect Productivity Loss among Academic Employees?

**DOI:** 10.3390/ijerph19020934

**Published:** 2022-01-14

**Authors:** Malin Lohela-Karlsson, Irene Jensen, Christina Björklund

**Affiliations:** 1Centre for Clinical Research, Region Västmanland–Uppsala University, Hospital of Västmanland, 721 89 Västerås, Sweden; 2Department of Public Health and Caring Sciences–Health Services Research, Uppsala University, P.O. Box 564, 751 22 Uppsala, Sweden; 3Centre for Musculoskeletal Research, Department of Occupational Health Sciences and Psychology, University of Gävle, 801 76 Gävle, Sweden; 4Unit of Intervention and Implementation Research on Worker Health (IIR), Institute of Environmental Medicine (IMM), Karolinska Institutet, 171 77 Stockholm, Sweden; irene.jensen@ki.se (I.J.); christina.bjorklund@ki.se (C.B.)

**Keywords:** commitment, job satisfaction, performance, productivity loss, work motivation, workplace

## Abstract

Work motivation and job attitudes are important for productivity levels among academic employees. In situations where employees perceive problems, for example, health-related and work environment-related problems, the ability to perform at work could be affected, which may result in fewer publications, reduced quality and less research funding. Few studies, however, have paid attention to productivity loss among academic employees in order to understand how, or if, the perceived loss is affected by the reported problems, either alone or in combination with work motivation and job attitudes. To evaluate whether attitudes towards work—measured as job satisfaction, organisational commitment and work motivation—are associated with productivity loss in the workplace, a cross-sectional study was conducted. This type of design is required as performance is highly variable and is affected by changes in health and work status. This study includes employees who reported either health-related problems, work environment problems or a combination of both (*n* = 1475). Linear regression analyses were used to answer the hypotheses. Higher levels of motivation, job satisfaction and organisational commitment were associated with lower levels of productivity loss among employees who experienced either health-related or work environment problems. High work motivation and high commitment were significantly associated with lower levels of productivity loss among employees who experienced a combination of problems. In summary, productivity loss among academic employees is not only affected by health-related problems or problems in the work environment but also by work motivation, job satisfaction and organisational commitment; i.e., these factors seem to buffer, or moderate, the reduction in performance levels for this group of employees.

## 1. Introduction

Getting published and attracting research funding is vital for the career advancement and prestige of academic employees. Scientific performance could differ between different areas of knowledge in the academy but often includes measures such as resources (e.g., funding, faculty support), the research process and output (e.g., publications, bibliometric indicators, teaching) [1,2,3]. Researchers have set out to identify the keys to success in academia, and several studies have found individual-, group- and organisational-level explanations [2]. Among other factors, differences in performance rates have been shown to be related to sex [4,5], family situation [6] and work tasks [7,8]. Several work environment-related and organisational-level factors have also been identified as important for high scientific performance; for example, highly competent, supportive leadership that emphasises scientific performance; a good communicative group climate and a transformative research group (i.e., new people are allowed to enter the group) [2]; and access to an external network and the necessary resources as well as clear goals and good communication [9]. In contrast, a deteriorating working environment, excessive workload and lack of institutional influence have been identified as barriers to scientific performance [10].

### 1.1. Work Attitudes and Motivation and the Impact on Performance among Academic Employees

Previous research has shown that job satisfaction, organisational commitment and work motivation are related to performance. Work motivation has been extensively studied within the field of organizational psychology and many different theories have been developed during the years [11]. Work motivation can be defined as a complex multidimensional determinant of direction and persistence in goal-directed behaviour [12]. One of the most well-articulated conceptual framework is the self-determination theory [13]. Self-determination theory stresses the importance of engaging in work activities for self-determined reasons, driven by pleasure, interest and personal importance [14]. The measure of work motivation used in this study is defined as willingness to work and was first introduced by Sjöberg and Lind (1994) [15]. Björklund (2001) [16] further developed this measure and related the measure to intrinsic motivation as well as factors such as work interest. Willingness to work was strongly related to intrinsic motivation as well as work interest. The measure willingness to work was built upon the notion that how willing a person is to work may be reflected in voluntarily actions. An argument for a volitional approach of measuring work motivation is that previous studies have shown that volitional approaches are the most efficient ones in predicting and explaining action [17,18].

Whereas work motivation may be described as the set of internal and external forces that promote work-related behaviour and determine its form, direction, intensity and duration [12], both job satisfaction and organisational commitment are better understood as job-related attitudinal constructs. Job satisfaction is unquestionably the job-related attitudinal construct that has received the most attention [19] and refers to the individual’s own assessment of his or her job in relation to issues that are important to the individual [20]. Organisational commitment, on the other hand, is defined as the bond employees experience in relation to the organisation, perceived costs associated with leaving the organisation and the obligation to remain with the organisation [21]. Researchers have given a great deal of attention to work motivation and attitudes towards work and the importance for scientific performance. In these studies, intrinsic motivation was shown to be important in explaining scientific performance [22,23,24,25,26,27]. For example, researchers who are motivated by internal self-concept (intrinsic motivation), i.e., an individual’s work motivation to adhere to internal standards in the field, will have a higher scientific performance [27,28]. Extrinsic motivation has been shown to be less important than intrinsic motivation for this group of employees [24,27,28]. It has also been suggested that employees who are more motivated by extrinsic factors, such as monetary rewards, will be less productive in their research [27]. Another study suggested different typologies of researchers with various patterns of intrinsic and extrinsic motivation and association with scientific productivity, which they mean indicated that academics have both extrinsic and intrinsic motives for undertaking research [29]. The relationship between research productivity and job satisfaction within academia has been extensively studied (e.g., [30,31]), and a positive correlation between productivity and job satisfaction has been reported [32]. Organisational commitment, on the other hand, has not been as strongly linked to productivity in this group. A recent study reported that organisational commitment was, in fact, positively correlated with productivity [33]. As has been shown in prior studies, scientific performance can be explained in part by work motivation and attitudes towards work.

### 1.2. Consequences of Poor Work Environment and Health-Related Problems on Performance

Previous studies have focused on what is produced or individual factors of importance for productivity without considering potential loss in individual performance or organisational productivity (see, for example, [1,2,23,27,28,32,33]). In situations where employees perceive problems, independent of related to the individual or the organisation, there is the potential that their performance will decrease. This means that the potential performance level is higher than the actual level, which could cause problems for the individual over the long run, as well as the larger research group, in terms of fewer publications, reduced quality and less research funding. It is therefore of interest to investigate whether perceived problems, independent of individual or organisational, result in decreased productivity and how this can be minimised.

Productivity loss can be defined as the difference between what employees normally produce and how much they produce when affected by the problem; i.e., if the employee perceives a problem that affect the employee’s performance at a lower level, a production loss will arise [34]. Most research studies have assessed health-related productivity loss, a measure that captures productivity loss due to health-related problems, whereas more recent research has introduced productivity loss due to work environment-related problems. Health-related productivity loss could be considered to capture a more individual perspective of the productivity loss while work environment-related productivity loss could be considered to capture an organizational perspective of productivity loss.

A few studies have investigated the impact of work environment-related and health-related problems on productivity loss in a university setting. For example, in a study investigating gender differences in perceived health-related problems and associated productivity loss, women reported a higher prevalence of health-related problems than men [35]. However, there were no gender differences in productivity loss for those who reported the problems. Another study conducted in a university setting investigated the association between work environment factors and health-related productivity loss and found that that the social climate affects the level of productivity loss among employees who report health-related problems [34]. These studies also investigated work environment-related problems and associated productivity loss. For example, the prevalence of perceiving a combination of both work environment-related and health-related problem was found higher among women than men [35]. No gender differences were found in the level of productivity loss among those who reported the problems. In a study investigating the association between work environment factors and related productivity loss, factors such as unfair leadership, inequality and role conflict are associated with productivity loss among academic employees perceiving work environment problems [36]. Despite the knowledge that work motivation and attitudes towards work are also important to the performance of academic employees, neither of the above-mentioned studies investigated whether these factors affected the level of productivity loss among employees who reported work environment-related and/or health-related problems.

Previous studies of academic employees have, to the best of our knowledge, looked at attitudes and work motivation separately from work environment factors and/or health problems, which fails to create an understanding of how, or if, these factors affect performance levels, either alone or in combination. Employees who are highly motivated could, for example, be less affected by a perceived work environment problem and therefore have lower levels of productivity loss than less motivated employees. It is also possible that employees who are satisfied with their work are less affected by perceived health problems than unsatisfied employees and therefore report lower levels of productivity loss. The same holds true for the level of commitment.

The aim of this study was to evaluate whether employee attitudes towards work and work motivation are associated with productivity loss. In this study, attitudes towards work are measured as job satisfaction and organisational commitment. Since job satisfaction, organisational commitment and work motivation have been positively correlated to job performance [37,38,39], we assume that all three factors could affect the level of productivity loss among employees who experience health-related problems or work environment-related problems. We assume that the performance levels of employees who are highly motivated, satisfied and committed to the organisation will be less affected than the performance levels of employees with low work motivation, low job satisfaction or/and low commitment, despite the presence of health- and/or work environment-related problems.

The hypotheses are:

Work motivation

**Hypothesis** **1a** **(H1a).**
*Employees with health-related problems who have high work motivation report lower levels of productivity loss compared to employees with lower motivation.*


**Hypothesis** **1b** **(H1b).**
*Employees with work environment-related problems who have high work motivation report lower levels of productivity loss compared to employees with lower motivation.*


**Hypothesis** **1c** **(H1c).**
*Employees with both work environment-related and health-related problems who have high work motivation report lower levels of productivity loss compared to employees with lower motivation.*


Job satisfaction

**Hypothesis** **H2a** **(H2a).**
*Employees with health-related problems who have high job satisfaction report lower levels of productivity loss compared to employees with low job satisfaction.*


**Hypothesis** **H2b** **(H2b).**
*Employees with work environment-related problems who have high job satisfaction report lower levels of productivity loss compared to employees with low job satisfaction.*


**Hypothesis** **H2c** **(H2c).**
*Employees with both work environment-related and health-related problems who have high job satisfaction report lower levels of productivity compared to employees with low job satisfaction.*


Commitment to the organisation

**Hypothesis** **H3a** **(H3a).**
*Employees with health-related problems who have a high commitment to the organisation report lower levels of productivity loss compared to employees with a low commitment to the organisation.*


**Hypothesis** **H3b** **(H3b).**
*Employees with work environment-related problems who have a high commitment to the organisation report lower levels of productivity loss compared to employees with a low commitment to the organisation.*


**Hypothesis** **H3c** **(H3c).**
*Employees with both work environment-related and health-related problems who have a high commitment to the organisation report lower levels of productivity loss compared to employees with a low commitment to the organisation.*


## 2. Materials and Methods

### 2.1. Participants

This cross-sectional study uses information collected through a questionnaire distributed in 2011. The questionnaire was part of a work environment survey at a medical university in Sweden and included questions about the psychosocial work environment, employee health and productivity loss.

All staff members who worked at least half-time and who had been employed for at least six months at the time of the screening were invited to participate (*n* = 5144). The invitation was sent by email to all staff meeting the inclusion criteria and was followed up by two reminders. Before participation, the employees were informed that the responses were anonymous for the employers. Participation in the study was encouraged but voluntary and each participating employee signed a written informed consent. A total of 3515 employees responded to the questionnaire (68 percent response rate).

This study included all employees who met the criteria for inclusion and reported that they had experienced health-related problems, work environment-related problems, or both, in the past seven days. These were identified using self-reported measures from the questionnaire. A total of 42 percent (*n* = 1475) of the respondents in the source population reported that they experienced health-related problems (*n* = 736), work environment-related problems (*n* = 307) or both work environment-related and health-related problems (*n* = 432) in the previous seven days, and were included in the study. Of the respondents who reported experiencing problems, 88 percent also reported that their problems affected their ability to perform while at work; i.e., a perceived productivity loss was reported. A description of the study population is presented in Table 1.

### 2.2. Measurements

The following two questions from the questionnaire were used to identify the employees that were included in the study population.

Health-related problems: To measure health-related problems at work, all employees were asked if they had experienced any health-related problems in the past seven days. In addition, a definition of health-related problems was provided and defined as any physical or psychological problems or symptoms. The response options were yes/no.

Work environment-related problems: Similar to the question on health-related problem, a question on work environment-related problems was used to identify those who had experienced any work environment-related problems in the past seven days. A definition of work environment-related problems was provided and defined as any physical, psychological or social problems that might arise in the work environment. Response options were yes/no.

Employees who responded yes on both health-related problems and work environment-related problems questions were included into the group of employees who experienced both type of problems. Employees who only reported health-related problems or work environment-related problems were divided into the groups “health-related problems” or “work environment-related problems”, respectively. Both questions have previously been used in other studies [34,40] and found important to use when studying productivity loss [40].

Work motivation and attitudes towards work were used as independent variables in the study.

Work motivation was in this study measured with a work motivation scale defined as willingness to work. Willingness to work can briefly be described as a person’s degree of willingness to continue a job task over a long time, even though it is not required, and if the person voluntarily puts more effort into his or her job. The measure used, a validated and reliable (α.70) [16] work motivation scale, was first introduced by Sjöberg and Lind [15], and has been used in other studies [41,42,43,44,45]. A short, validated and reliable version of the scale was developed [16] and has been used in several studies [44,45,46,47]. In this study, a similar short version of the work motivation scale was used. The following questions were included: “Do you feel stimulated by your work tasks?”; “Are you motivated to work?”; “How often do you feel a strong will to work?”; and “Would you spend less time at work if possible?”. The items were evaluated on a five-point scale, ranging from 5 = always to 1 = never. A motivation index was summarised from the single items’ response options, ranging from 1 to 5, with a higher score indicating higher motivation.

Attitudes towards work was measured as job satisfaction and organizational commitment.

Job satisfaction. Most often, job satisfaction is conceptualised as a general attitude towards an object. Job satisfaction was measured with a single question designed to capture a respondent’s general attitude towards work [16]. Several articles have discussed whether a single-item measure is adequate to measure overall job satisfaction. The conclusion drawn in most studies is that a single-item measure of overall job satisfaction is acceptable [48,49]. The items were evaluated on a five-point scale, ranging from 5 = very satisfied to 1 = very unsatisfied.

Organisational commitment was assessed using the validated questionnaire General Nordic Questionnaire for Psychological and Social Factors at Work (QPS Nordic) [50]. The response options ranged from “very seldom or never” to “very often or always”. The items were summed up and standardised to generate a total score ranging from 1 to 5, where a high score is favourable.

Two self-rated measures of productivity loss were used as dependent variables in the study.

Health-related productivity loss: Those who responded yes to the question on health-related problems were asked if their health problems had affected their performance at work in the past seven days. Health-related productivity loss was measured using the validated question health-related productivity loss (HRPL) [40,51]. The question was based on one of the items in a previous questionnaire [52] and phrased in the following way “During the past seven days, how much did your health problems affect your performance while you were working? Think about days you were limited in the amount or kind of work you could do, days you accomplished less than you would like, or days you could not do your work as carefully as usual. If health problems affected your work only a little, choose a low number. Choose a high number if health problems affected your work a great deal”. Response options ranged from 0 to 10, where 0 = “Health problems had no effect on my work” and 10 = “Health problems completely prevented me from working”.

Work environment-related productivity loss: Those who reported work environment-related problems were asked to answer the validated question work environment-related productivity loss (WRPL) [34,40], which captures the effect of work environment-related problems on reduction in work performance. Similar to the question on HRPL, all respondents were asked to quantify how much work environment problems had affected their performance at work in the past seven days. Work environment problems were, similarly as for the question on work environment-related problems, defined as any physical, psychological or social problems that resulted from the work environment. The response scale ranged from 0 to 10, where 0 = “Work environment problems had no effect on my work” and 10 = “Work environment problems completely prevented me from working”.

### 2.3. Statistical Analyses

Before performing the statistical analyses, the work attitude factors, i.e., the predictors, were categorised into high and low, respectively, using the median. The category “low” was defined as all values below the median, and the category “high” was defined as the median value and all values above. The median values were based on the population with the specific problem (see Table 1). For example, the median value of work motivation for employees with health-related problems was used to categorise the variable included in that regression analysis. The decision to categorise the predictors was based on the non-normal distribution of the variables.

In order to answer the hypotheses, a linear regression analysis was performed for each outcome variable, namely, health-related productivity loss, work environment-related productivity loss as well as productivity loss due to both work environment and health-related problems, with its hypothesised predictor. The combined outcome variable was calculated as the average productivity loss of both health-related and work environment-related productivity loss for those reporting that they had experienced both problems in the previous seven days. This decision to use linear regression was based on that the dependent variables were approximal normally distributed. The model was then controlled for using the following covariates: sex, age and occupation (Model 2).

To test for the sensitivity of the results depending on the chosen cut-offs for the predictors, the median value of the whole source population’s work attitudes was used as the cut-off. Health-related productivity loss was not normally distributed in this population, which is probably due to the fact that this is a working population. Therefore, as an additional test, health-related productivity loss was categorised into high versus low productivity loss using the median value, and a general estimated equation was performed.

## 3. Results

The median work motivation was highest in the overall population and lowest among those experiencing both health-related and work environment-related problems. Organisational commitment was somewhat lower among those who reported experiencing work environment-related problems or a combination of problems compared to the overall population and those experiencing health-related problems only. Job satisfaction was equal for all groups.

Table 2 presents the average productivity loss for each group divided into the different levels of motivation, job satisfaction and commitment. This indicates that the average productivity loss is lower among employees who reported higher levels of motivation, job satisfaction and commitment.

### 3.1. Work Motivation

The results from the regression analyses show that employees who experienced health-related problems and were highly motivated had, on average, lower levels of productivity loss than employees that were less motivated (Table 3, Model 1). These associations remain even after including the covariates (Table 3, Model 2). Employees who experienced work environment-related problems and who were highly motivated also had lower levels of productivity loss than those who were less motivated. The same association was found among employees with a combination of problems; i.e., the average productivity loss was lower among those with higher work motivation than among those with lower work motivation. These results support Hypotheses H1a–H1c.

### 3.2. Job Satisfaction

The level of productivity loss was lower for employees reporting higher levels of job satisfaction compared to those reporting lower levels of job satisfaction. This association was significant for both the group who experienced health-related problems and for those experienced work environment-related problems. This association, however, was not significant for those who experienced a combination of problems. These results support Hypothesis H2a and Hypothesis H2b but not Hypothesis H2c.

### 3.3. Commitment to the Organisation

Employees with health-related problems who were more committed to the organisation reported lower levels of productivity loss compared to employees who were less committed to the organisation. The same association was found for employees who experienced work environment-related problems and for those who experienced a combination of problems. The significance remains even though the covariates are included in the regression analyses. These results are in accordance with Hypotheses H3a–H3c.

### 3.4. Sensitivity Analysis

A sensitivity analysis was performed using the median values of the total population as cut-offs for the predictors, in addition to the health- or work environment-related population specific cut-offs (data not presented). This change did not result in any large differences in the results, other than a change in the number of employees in each category, which sometimes reduced in number. As indicated above, health-related productivity loss was not normally distributed in this population. Therefore, health-related productivity loss was categorised into high versus low productivity loss, using the median value to evaluate for any differences in the results. No change was found; i.e., the result regarding the significance of work attitudes and motivation remained the same as when health-related productivity loss was used as a continuous variable.

### 3.5. High/Low Levels of Motivation/Attitudes and High/Low Levels of Productivity Loss

The employees were categorised into four different groups depending on their level of motivation/attitude and level of productivity loss. This was done separately for the group with health-related problems, the group with work environment-related problems and the group with a combination of problems (Table 4). According to the results presented above, employees with health-related problems or work environment-related problems, who also report higher levels of work motivation, job satisfaction and commitment, have lower levels of productivity loss. For employees with a combination of problems, who also report higher levels of work motivation or commitment, the levels of productivity loss are lower. The group with a combination of problems had the largest proportion of employees with low work motivation and high productivity loss, whereas the group with employees who experienced only health-related problems had the smallest proportion of employees. The same pattern is also seen for job satisfaction and organisational commitment.

## 4. Discussion

This study investigated the relationship between work motivation, work attitudes and self-reported productivity loss among academic employees. This relationship was studied for a group of employees that reported experiencing health-related problems, work environment-related problems or a combination of both. The overall analyses showed that higher levels of work motivation, organisational commitment and job satisfaction were associated with lower levels of productivity loss among employees who experienced either health-related or work environment-related problems. For the group of employees who experienced a combination of problems, the results showed that high work motivation and high organisational commitment were significantly associated with lower levels of productivity loss.

### Work Attitudes, Motivation and Productivity Loss

Work motivation has previously been shown to be an important factor for employee performance levels [37], where employees who have intrinsically high levels of motivation are more likely to engage in a task and to put more effort into the task [13]. This was also found to be an important explanation for scientific performance [22,23,24,25,26,27]. In this paper, we also study a different perspective of performance: productivity loss. Similar to the findings in the psychology research, we show that work motivation is an important factor for the level of productivity loss among academic employees, regardless of the cause of the problem (ill-health and work related). Highly motivated employees report lower levels of productivity loss, which could be interpreted as that motivation can moderate the consequences of ill-health and work environment problems on the ability to produce at work, or function as a buffering factor, and contribute to maintaining a high(er) performance level, at least over the short term. Being present at work while experiencing health-related problems has previously been associated with an increased risk of future ill-health [53,54]. How work motivation affects academic employees who experience health or work environment-related problems, and the impact on performance levels and future health, is still unknown. In a study of working adults, high work motivation has in itself been identified as a risk factor for exhaustion [46]. It would be interesting to design a more in-depth study of how work motivation in a group of academic employees who experience health- or work environment-related problems affects health and performance, or productivity loss, over an extended period of time. Mobility among researchers is limited to a certain extent, and it can be difficult to find job openings in academia, which might cause employees to remain in their position even when they experience these types of problems at their current workplace. Intrinsic motivation could function as a buffering strategy to prevent future ill-health, sick leave or undesired turnover.

In this study, lower levels of productivity loss were reported among those who were highly satisfied compared to those less satisfied. A previous study showed that employees who were satisfied with their job or the work that they did reported a higher job performance compared to those that were dissatisfied [55]. The study population in the Merrill et al. study [55] consisted of a broad population of employees, working in several different companies. In the present study, we only included employees with health-related and/or work environment-related problems, as the outcome of interest was measured in terms of productivity loss due to either of these problems. The study population also differed. Despite the differences in the population, we found a similar pattern to the Merrill et al. study; that is, employees who reported either health-related or work environment-related problems, who were highly satisfied with their job, also reported significantly lower levels of productivity loss. This association was not significant for the group with a combination of problems even though the trend was similar. However, the number of employees who reported high levels of job satisfaction in the group with a combination of problems was relatively small, and a larger sample size may have increased the power.

The level of productivity loss was also significantly related to organisational commitment for the studied population. However, the results show that the difference in the level of productivity loss between those who were highly committed compared to those who were less committed was higher in the groups who reported work environment-related problems than those reporting health-related problems. This indicate that organisational commitment explains productivity loss due to work environment problems to a greater extent than work motivation. Organisational commitment has previously been shown to be related to job performance; however, in several of the meta-analyses, the relationship between organisational commitment and performance was found to be weak [56] and mixed [57] while work motivation was shown to be more strongly correlated to performance [37]. This could be interpreted as that motivation and commitment capture different dimensions of performance, where motivation is more related to overall performance and commitment is related to productivity loss due to work environment problems. As in the case of organisational commitment, job satisfaction also explains productivity loss to a higher extent than work motivation. However, job satisfaction is more strongly correlated with health-related productivity loss, whereas organisational commitment is more strongly correlated with productivity loss due to work environment problems. The results for organisational commitment could also be related to the type of organisation in which the respondents are employed. The organisation might have a good reputation, which increases employee pride; i.e., the results are more related to the organisation in itself, not to commitment in general.

## 5. Conclusions

Productivity losses are not only affected by health-related problems or problems in the work environment but also by employee motivation, job satisfaction and organisational commitment. Higher levels of work motivation, organisational commitment and job satisfaction were found associated with lower levels of productivity loss among employees who experienced either health-related or work environment-related problems. Work attitudes and work motivation seem to buffer, or moderate, the reduction in performance levels of academic employees who experience these types of problems. Employers could potentially decrease the level of productivity loss by conducting activities that increase employee motivation or improve attitudes towards work, at least in the short run.

This study is a cross-sectional study, which, depending on the topic studied, can present limitations. Longitudinal studies are used more frequently in contemporary research. However, our research questions in this study, which aimed to investigate the relationship between work attitudes and performance through a self-rated performance measure, require a cross-sectional study design inasmuch as performance is highly variable and is affected by current health status and work environment conditions. Thus, this design is essential to capture health status, present work environment conditions and performance at the same point in time.

One limitation of this study is the lack of objective data measuring performance. Further studies are recommended to test this study’s hypotheses using objective data, in addition to the subjective measure of productivity loss, to evaluate whether employees who perceive problems with corresponding productivity loss also produce less in objective terms. This should especially be investigated in relation to attitudes and work motivation to determine whether the results found in this study are valid when objective measures of performance are used.

## Figures and Tables

**Table 1 ijerph-19-00934-t001:** Description of the study population.

	Total Population (*n* = 3515)	Health-Related Problems (*n* = 736)	Work Environment Problems (*n* = 307)	Both Health and Work Environment Problems (*n* = 432)
Total, *n* (%)	3515	736 (21)	307 (9)	432 (12)
Gender, *n* (%)				
Men	1190	222 (19)	87 (7)	97 (8)
Women	2325	514 (22)	220 (9)	335 (14)
Age, mean (sd)				
*n* (%)	43 (12)	43 (12)	43 (11)	42 (11)
≤29	463	106 (23)	41 (9)	58 (13)
30–39	1096	231 (21)	98 (9)	144 (13)
40–49	796	172 (22)	71 (9)	103 (13)
50–59	710	129 (18)	65 (9)	100 (14)
≥60	450	98 (22)	32 (7)	27 (6)
Manager, *n* (%)				
Yes	655	129 (20)	62 (9)	67 (10)
No	2860	607 (21)	245 (9)	365 (13)
Occupation, *n* (%)				
Researcher	2022	433 (20)	170 (8)	216 (11)
Administrative staff	1493	303 (21)	137 (9)	216 (14)
Years at current workplace, *n* (%)				
<1 year	270	46 (17)	22 (8)	38 (14)
1–2 years	709	163 (23)	68 (10)	93 (13)
3–5 years	954	202 (21)	85 (9)	120 (13)
6–10 years	582	137 (24)	53 (9)	66 (11)
>10 years	1000	188 (19)	79 (8)	115 (12)
Work motivation, Median (var)	4.67 (0.61)	4.33 (0.61)	4.33 (0.77)	4.00 (0.93)
Organizational commitment, Median (var)	3.67 (0.79)	3.67 (0.69)	3.33 (0.82)	3.33 (0.92)
Job satisfaction, Median (var)	4.00 (0.82)	4.00 (0.87)	4.00 (1.02)	4.00 (1.25)
Production loss due to the problem, *n* (%)				
Yes	-	609 (83)	274 (91)	418 (97)
No	-	127 (17)	28 (9)	14 (3)

**Table 2 ijerph-19-00934-t002:** Proportion of employees with high/low work motivation, job satisfaction and organisational commitment towards work presented for the groups with different problems, as well as the average productivity loss divided by the level of work attitudes in each group.

	Health-Related Problems		Work Environment Problems		Both Health and Work Environment Problems	
	(*n* = 736)		(*n* = 307)		(*n* = 432)	
	*N* (%)	Average Production Loss	*N* (%)	Average Production Loss	*N* (%)	Average Production Loss
Work motivation						
High	452 (61)	2.72	180 (59)	3.46	189 (44)	3.61
Low	284 (39)	3.68	122 (41)	4.27	242 (56)	4.40
Job satisfaction						
High	189 (26)	2.26	50 (16)	2.96	54 (13)	3.57
Low	547 (74)	3.37	252 (84)	3.95	377 (87)	4.12
Commitment						
High	302 (41)	2.68	174 (57)	3.38	222 (51)	3.55
Low	434 (59)	3.37	128 (43)	4.34	209 (49)	4.59

**Table 3 ijerph-19-00934-t003:** Association between work motivation, job satisfaction and organisational commitment and productivity loss among employees who experience health-related problems, work environment-related problems or a combination of problems.

	Health-Related Problems *n* = 736		Work Environment-Related Problems *n* = 307		Both Health- and Work Environment-Related Problems *n* = 432	
	β	CI	β	CI	β	CI
Model 1	Adj R2 = 0.029		Adj R2 = 0.022		Adj R2 = 0.027	
Work motivation						
Low	0.955	0.560–1.349	0.809	0.234–1.385	0.799	0.366–1.232
High	0		0		0	
Model 2 ^1^	Adj R2 = 0.036		Adj R2 = 0.046		Adj R2 = 0.028	
Work motivation						
Low	0.867	0.462–1.272	0.876	0.297–1.454	0.825	0.386–1.264
High	0		0		0	
Model 1	Adj R2 = 0.031		Adj R2 = 0.018		Adj R2 = 0.004	
Job satisfaction						
Low	1.110	0.671–1.549	0.992	0.231–1.753	0.549	−0.108–1.207
High	0		0		0	
Model 2 ^1^	Adj R2 = 0.041		Adj R2 = 0.034		Adj R2 = 0.003	
Job satisfaction						
Low	1.034	0.594–1.473	0.871	0.108–1.633	0.541	−0.119–1.201
High	0		0		0	
Model 1	Adj R2 = 0.015		Adj R2 = 0.033		Adj R2 = 0.049	
Commitment						
Low	0.691	0.298–1.084	0.964	0.396–1.533	1.037	0.611–1.462
High	0		0			
Model 2 ^1^	Adj R2 = 0.028		Adj R2 = 0.056		Adj R2 = 0.050	
Commitment						
Low	0.669	0.277–1.060	0.996	0.434–1.558	1.060	0.633–1.486
High	0				0	

^1^ Controlled for the covariates sex, age and occupation.

**Table 4 ijerph-19-00934-t004:** Proportion of employees with high/low attitudes towards work and high/low levels of production loss, presented for the groups with different problems.

	Health-Related Problems (*n* = 736)		Work Environment Problems (*n* = 307)		Both Health and Work Environment Problems (*n* = 432)	
	Production Loss		Production Loss		Production Loss	
*N* (%)	High	Low	High	Low	High	Low
Work motivation						
High	121 (16)	331 (45)	73 (24)	107 (35)	76 (17)	113 (26)
Low	119 (16)	165 (23)	68 (23)	54 (18)	136 (32)	106 (25)
Job satisfaction						
High	41 (6)	148 (20)	17 (6)	33 (11)	24 (6)	30 (7)
Low	199 (27)	348 (47)	124 (41)	128 (42)	188 (44)	189 (44)
Commitment						
High	79 (11)	223 (30)	69 (28)	105 (43)	86 (20)	136 (32)
Low	161 (22)	273 (37)	72 (29)	0 (0)	126 (29)	83 (19)

## Data Availability

The data presented in this study are available on request from the corresponding author. The data are not publicly available due to privacy reasons.

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
