# Peer review of "Do Attitudes towards Work or Work Motivation Affect Productivity Loss among Academic Employees?"

_ijerph, 2022, doi:10.3390/ijerph19020934_

Round 1
Reviewer 1 Report
The abstract does not clearly identify the objective of the work and the methodology used. The same happens in the introduction, where the key factors for the study are also explained with some citations, but I see it necessary to increase this section with more and above all more current citations.
In the sections where the literature review takes place, the bibliography is again short and out of date. For example, in lines 76 to 79, previous or anticipated studies are mentioned, but the statements given are not cited and the statements given are not justified, the same occurs in lines 102 to 109. Therefore, the hypotheses are not fully justified with the revision of the literature made.Regarding the methodology section, the date of launch of the questionnaire is striking, in addition to the fact that health problems are defined without justification or citation in the majority.There is talk of linear regression as the methodology used, for the first time, but it is not explained why that and not another.The conclusions should be expanded.Author Response
The authors are very thankful for the reviewers comment to our manuscript and appriciate the time used to share their experise which we feel have helped us approve our work. The comments have been responded to point-by-point, please see the attachment.

Reviewer 2 Report
The manuscript deals with the impact of attitudes towards work (job satisfaction, organisational commitment and work motivation), as well as work environment problems and health-related problems on the scientific performance of academic employees.
It is noted that the overall approach and development of the manuscript is adequate although some aspects should be improved.
Introduction.
In the introduction, it is stated that sex, family situation and work tasks affect job performance.
The sex variable is not studied extensively in this research. It should be justified why it has not been studied.
The variable age is used in the statistical Analyses but is not addressed in the literature review. This section should be developed in the theoretical framework.
The occupation variable is considered in the statistical analysis. As with the other variables mentioned above, it has not been developed in-depth in the theoretical framework.
The motivation variable is developed in a very superficial way. There are different theories (achievement goal theory, self-determination theory...) and theoretical approaches that could have been considered. This section should be improved.
The variable experience at current work is considered in the statistical analysis but not in the theoretical framework.
Point 1.2 deals at the same time with the variables of poor work environment and health-related problems. It is suggested that the authors explain each of these two factors in more depth, following the same discourse. In other words, if the gender perspective is discussed in one factor, this aspect should also be developed in the other.
The concept of health is poorly explained and it should be indicated that it integrates aspects associated with physical, psychological and social components of well-being. Somewhere in the text, it should be justified why three groups of participants with health problems related to these three health domains have not been considered.
The concept of work environment problems is defined in the same way as health problems. This section may be directly related to the type of task or role at work. It should be further explained in the theoretical framework. It will certainly have a direct bearing on the dimensions or factors considered in the theoretical framework that originated the WRPL questionnaire.
When mentioning questionnaires, it is recommended to put the full name of the questionnaire and in brackets the acronym (HRPL, WRPL, QPS).
Regarding the Work Motivation questionnaire, the authors indicate that a reduced scale was used. It should be explained how the original scale of attitudes towards work and work motivation with 12 questions was changed to the short version with 4 questions. How these four questions were identified. What statistical tests were carried out to decide on the four items.
Apartado 2. Materials and Methods
You should introduce the section:
2.1 Participants: describe the sample very well: n, age range, sex and SD.
Given the nature of this research, the authors should indicate the code number of the approval of the study by an official ethics committee.
2.1 Material and data collection
The content of this section could be included in the participants' section.
2.2 Measurements.
Scientific output or productivity should be included as it is the main dependent variable of this study.
In the theoretical framework, it would be convenient to develop in more depth what it means to talk about depth in different areas of knowledge (number of publications, type of publications, participation in scientific projects, elaboration of dissemination materials, transfer in degree courses, PhD or Master's programmes...).
Verb tenses.
When describing the objectives, hypotheses and also in the first paragraph of the discussion, it is recommended to use the simple past tense.
Discussion
Aspects referring to limitations or future perspectives should be placed at the end of the discussion or in a section integrating conclusions, limitations and future perspectives (e.g. text lines 368-371).
Change the name Methodological considerations to limitations. Can be merged in the conclusions section.
Conclusions.
This section should contain the most important ideas about all the variables considered in the study, which have been statistically analysed and which should have been developed in the discussion.
Informed Consent Statement. The code of approval of the research by an official ethics committee should be included.
Author Response
The authors are very thankful for the reviewers comment to our manuscript and appriciate the time used to share their experise which we feel have helped us approve our work. The comments have been responded to point-by-point, please see the attachment.

Round 2
Reviewer 1 Report
The authors have made all the changes suggested by me. So I think the article is ready to be published.Reviewer 2 Report
Dear Authors,
Congratulations on the efforts you have done to improve your manuscript.